# Longitudinal changes in IgG levels among COVID-19 recovered patients: A prospective cohort study

**Ashraf Hassan Alzaabi**[1]*, **Luai A. Ahmed**[2,3], **Abdulla E. Rabooy**[4], **Ali Al Zaabi**[5], **Mohammed Alkaabi**[6], **Falah AlMahmoud**[4], **Mai Farouk Hamed**[7], **Khalid Omar Bashaeb**[4], **Abdul Rahim Bakhsh**[8], **Suha Adil**[8], **Nadeen Elmajed**[8], **Ahmed Nigm Abousalha**[8], **Ahmad Kanaan Uwaydah**[9], **Khulood Al Mazrouei**[1]

1 Pulmonology Department, Zayed Military Hospital, Abu Dhabi, United Arab Emirates, 2 Institute of Public Health, College of Medicine and Health Sciences, United Arab Emirates University, Al Ain, United Arab Emirates, 3 Zayed Centre for Health Sciences, United Arab Emirates University, Al Ain, United Arab Emirates, 4 Radiology Department, Zayed Military Hospital, Abu Dhabi, United Arab Emirates, 5 Pathology and laboratory Department, Zayed Military Hospital, Abu Dhabi, United Arab Emirates, 6 Microbiology Department, Zayed Military Hospital, Abu Dhabi, United Arab Emirates, 7 Laboratory Department, Zayed Military Hospital, Abu Dhabi, United Arab Emirates, 8 Internal Medicine Department, Zayed Military Hospital, Abu Dhabi, United Arab Emirates, 9 Infectious diseases Department, Zayed Military Hospital, Abu Dhabi, United Arab Emirates

* ashrafalzaabi@hotmail.com

**Data Availability Statement:** All the relevant data are within the manuscript. The complete dataset is not publicly available due to ethical restrictions as data contain potentially sensitive information. Data

## Abstract

### Objectives

To quantify SARS-CoV2 IgG antibody titers over time and assess the longevity of the immune response in a multi-ethnic population setting.

### Setting

This prospective study was conducted in a tertiary hospital in Abu Dhabi city, UAE, among COVID-19 confirmed patients. The virus-specific IgG were measured quantitatively in serum samples from the patients during three visits over a period of 6 months. Serum IgG levels ≥15 AU/ml was used to define a positive response.

### Participants

113 patients were analyzed at first visit, with a mean (SD) age of participants of 45.9 (11.8) years 87.5% of the patients were men. 63 and 27 participants had data available for visits 2 and 3, respectively.

### Primary outcome

Change in SARS-CoV2 IgG antibody titers over the visits.

### Results

No mortality or re-infection were reported. 69% of the patients developed positive IgG response within the first month after the onset of symptoms. The levels of IgG showed a

would be available upon justified request after approval of the Research Ethics Committee at Zayed military Hospital. The contact person for requesting the data would be Dr. Jumaa Aldhaheri who is the Chief Medical Officer. His contact info is as follows: Dr. Jumaa Aldhaheri Chief Medical Officer Zayed Military Hospital, Medical Services Corps P.O. Box 3740 Abu Dhabi, U.A.E. cmo@msc.mil.ae Tel:+971509966780 (Direct) Tel: +97124055000.

**Funding:** This study was funded by unrestricted educational grant from Zayed military hospital (No grant number).

**Competing interests:** The authors declare no competing interests.

consistent increase during the first three months with a peak level during the third month. Increasing trend in the levels of IgG were observed in 82.5%, 55.6% and 70.4% of patients between visit 1 to visit 2, visit 2 to visit 3, and from visit 1 to visit 3, respectively. Furthermore, about 64.3% of the patients showed sustained increase in IgG response for more than 120 days.

## Conclusions

Our study indicates a sustained and prolonged positive immune response in COVID-19 recovered patients. The consistent rise in antibody and positive levels of IgG titers within the first 5 months suggest that immunization is possible, and the chances of reinfection minimal.

## Introduction

The pandemic of the severe acute respiratory syndrome coronavirus 2 (SARS-CoV2) RNA viral infection that causes the highly contagious coronavirus disease 2019 (COVID-19) has resulted in severe morbidity and mortality in humans [1].

Antibody response is one of the key factors for development of immunity and preventing re-infection. In general, following infection, the body's immune system blocks the viral entry with neutralizing natural antibodies [2–5]. During SARS-CoV2 viral infection, the fusion process of the virus with the host cell membranes is facilitated by two spike (S) glycoproteins S1 and S2 [6]. These proteins are significant immune targets for the development of specific antibodies. Monitoring this antibody response is an important aspect of understanding the scope of immunity development. It is also one of the key factors for a successful vaccine development.

As the vaccination against COVID-19 is under development and is not yet available for wide use, there is an increasing interest in investigating the persistence of antibody response against COVID-19 virus [2,5,7–11]. The humoral response a few weeks after SARS-CoV-2 infection has been thoroughly described, some studies reported stable antibody levels within the first three months of recovery, whereas others showed a rapid decrease in convalescent patients [12–16].

However, data on the evolution of antibody levels beyond the third month is limited and only few studies have recently documented it, suggesting a stable antibody immunity six months after the SARS-CoV2 infection [17–20,21]. These studies mostly came from Europe and USA and it is essential to have data from all over the world, with multi-ethnic participants. Up to now, there is no evidence of protection against second infection and it is essential to continue assessing the antibody response for a prolonged period in COVID-19 patients [5,12].

In this backdrop, this prospective study has followed COVID-19 patients for nearly six months, to quantify IgG antibody titers over time and assess the longevity of the immune response in a multi-ethnic population setting.

## Methods

### Study design and setting

This prospective study was conducted in a military tertiary hospital in Abu Dhabi city, UAE. All COVID-19 confirmed patients admitted between March and May 2020 were eligible to be included in the study, being symptomatic or asymptomatic at the time of admission. The first

patients arriving and willing to participate during this period were included. Patients were not involved in the design, or conduct, or reporting, or dissemination plans of the study research.

Demographic, previous medical history, clinical presentations and characteristics, chest computed tomography (CT) findings, laboratory tests, treatment and outcome data were all maintained as inpatient medical records. Data regarding age, gender, nationality, smoking history, existence of comorbidities such as diabetes, hypertension, cardiovascular diseases, renal diseases, immunodeficiency syndrome, autoimmune disorders and other respiratory diseases was collected by the physician. Detailed history was obtained that included presence of following symptoms: fever, cough, shortness of breath, fatigue, loss of taste, loss of smell, sore throat, nausea, diarrhea or any other symptoms informed by the patient. Laboratory data collected for each patient included complete blood count, coagulation profile, serum biochemical tests (including renal and liver function, lactate dehydrogenase and creatine kinase), serum ferritin and other biomarkers of infection. Chest CT scans were done for all inpatients.

Three repeated measurements of IgG titers were initially planned: the first one during the hospitalization, the second one a few weeks after the first symptoms and a third one a few months after the first symptoms. Measurements of IgG titers were performed on serum samples obtained from the patients during hospitalization and/or subsequent recalls to visit the hospital to perform the tests over a period up to 6 months after the onset of symptoms.

The study was approved by the research Ethics Committee at Zayed military hospital (Ethics approval ID 2020.6) and all participants provided written informed consent before inclusion.

## Patient and public involvement

No patient involved.

## Data sources/measurement

**RT-PCR.** Nasopharyngeal swabs were collected at least twice and a qualitative detection of SARS-CoV2 virus was performed using real-time RT–PCR. Viral ribonucleic acid (RNA) from all samples were isolated within 24 hours and according to manufacturer instructions using the commercial RT–PCR kit. The novel coronavirus (2019-nCoV diagnostic kit (PCR-Fluorescence Probing) was used for testing the samples. This kit is used for detecting two target genes: open reading frame1ab (ORF1ab) and nucleocapsid protein (N) genes, which were simultaneously amplified and tested. The PCR machine used is of the model MA6000 Real-time quantitative PCR system.

**CT-scan.** All the patients were investigated with CT examination of the chest to look for features of COVID-19 pneumonia, including lower lobe predominant, peripherally dominant, multiple, bilateral ground-glass opacities (GGO) with or without crazy paving, peripheral consolidation, air bronchograms and reverse halo/perilobular patterns. The examination was performed on a 64 multislice CT scanner using volumetric scanning with 1 mm multiplanar high resolution lung and soft tissue reconstructions. The CT images were then independently reviewed by three (3) consultant radiologists for changes of COVID-19 pneumonia. The findings of the CT examination of each patient were then given two scores. The first according to the British Society of Thoracic Imaging (BSTI) classification for COVID-19 infection and the second an overall imaging opinion of the presence or absence of infection with a 'Yes', or 'No' score. The BSTI pattern score is a five-point scoring system on patterns of imaging appearances with varying confidence of the presence of COVID-19 pneumonia (S1 Table in S1 File). The results were reviewed and the cases where there was lack of agreement, consensus agreement was reached.

**Serology tests.** The acute antibody response to SARS-CoV-2 infection, virus-specific IgG were measured quantitatively in serum samples from the patients. The assay for quantitative measurement of SARS CoV-2 IgG against S1 and S2 viral spike proteins was performed using Liaison XL SARS-CoV-2 S1/S2 IgG reagent which uses Chemiluminescence immunoassay technology according to DiaSorin's instructions. The IgG titers were measured in arbitrary units per milliliter (AU/ml). More details are available in the supplementary material.

## Variables

A confirmed case of COVID-19 was defined as an individual with nasopharyngeal swabs that were positive for SARS-CoV-2, tested using laboratory-based reverse transcriptase polymerase chain reaction (RT-PCR).

A symptomatic patient was defined as a patient with one or more of the COVID-19 related symptoms. An asymptomatic patient was defined as an individual who tested positive for COVID-19 but without any relevant clinical symptoms at the time of hospitalization.

The date of admission and the date of noticing the first symptoms were recorded. All the time durations in this study are measured from the onset of symptoms in symptomatic individuals. In the asymptomatic patients, the results of a positive COVID-19 test date were considered as the first day.

## Bias

Due to implementation of lockdown and access restrictions measures in the city during the study period, complete follow-up of serological investigations in all patients was not feasible. To address this potential selection bias, comparison of baseline characteristics between those having a complete follow-up (three IgG levels data) and those who had only the first IgG available was done. Also, the main analysis comparing IgG levels at the three visits was performed only among those with complete follow-up.

## Quantitative variables

The IgG values between 0–12 AU/ml, 12–14·99 AU/ml and equal to or more than 15 AU/ml were considered as negative, equivocal, and positive, respectively. Equivocal IgG values were interpreted as negative. More details are available in the supplementary material.

## Statistical analysis

Descriptive statistics were performed to present and compare the distribution of the patients' demographic and clinical characteristics according to the IgG response status. Continuous variables were described using mean and standard deviation (SD), while categorical variables were described using counts and percentages. IgG levels were presented using median and interquartile range (IQR). The differences between groups were examined using Welch's t-test and Mann-Whitney U test for continuous variables with skewed distribution and Fisher's exact test for categorical variables. Logarithmic transformation of the IgG values was used to present the distributions during the visits and to illustrate the increasing and decreasing trends in IgG levels among the patients between the visits. To report the monthly variations of the IgG levels, the distribution of all the available measurements within every 30-days intervals after the start of the symptoms was reported. As results may differ between symptomatic and asymptomatic patients, and according to the sex, two sensitivities analyses were done: according to presence of symptoms and according to sex.

Logistic regression was used to test the association between several factors (age, nationality, symptoms, presence of comorbidity, viral shedding, treatment duration, and Biochemical parameters) and an increase in IgG response from first to second visit, first in univariate model and then adjusting on age.

The analyses were performed using Stata 16·1 (Stata Corp, College Station, TX) and graphs were made using GraphPad Prism version 8·4·3.

## Results

Among the 113 patients with serology tests, 63 patients managed to attend the second visit, and 27 patients provided three repeated IgG measurements, as shown by the flowchart representation of the study design in Fig 1.

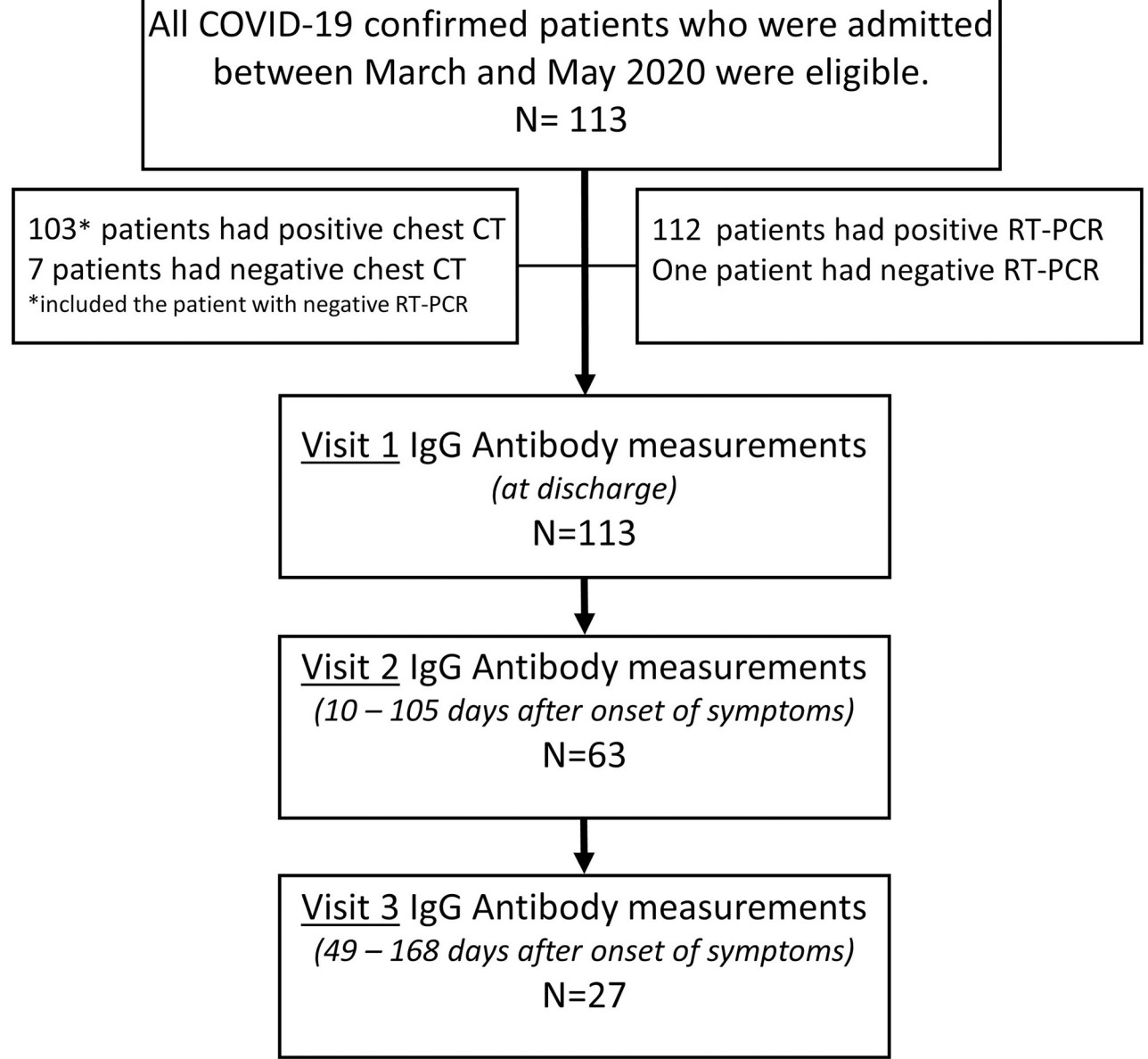

**Fig 1. Study design.**

**Table 1. Baseline demographics, clinical and biochemical characteristics (N = 113).**

|  | N(%) |
|---|---|
| **Sex** |  |
| Male n (%) | 99 (87.6%) |
| Female (n, %) | 14 (12.4%) |
| **Age in years,** *mean ± SD* | 45.9 ± 11.8 |
| **Age groups** |  |
| 20–29 | 9 (8.0) |
| 30–39 | 29 (25.7) |
| 40–49 | 28 (24.8) |
| 50–59 | 30 (26.6) |
| 60+ | 17 (15.0) |
| **Ethnicity** |  |
| UAE National n (%) | 42 (37.5%) |
| International n (%) | 70 (62.5%) |
| **Symptomatic status** |  |
| Symptomatic patients, n (%) | 89 (78.8%) |
| Asymptomatic patients, n (%) | 24 (21.2%) |
| **Presence of comorbidity**[*] |  |
| **Yes** | 61 (54.0) |
| **No** | 52 (46.0) |
| **Biochemical parameters collected on the day of admission,** *mean ± SD* |  |
| **WBC** | 6.17 ± 2.59 |
| Absolute lymphocyte count | 1.67 ± 0.74 |
| Lactate dehydrogenase | 227 ± 85.9 |
| D-Dimer | 0.437 ± 0.448 |
| Ferritin | 365 ± 524 |
| C-Reactive protein (CRP) | 18.4 ± 32.1 |
| Creatinine protein kinase (CPK) | 180 ±237 |

SD- Standard deviation,

[*]Includes: Diabetes Mellitus, Hypertension, Heart Disease, Renal Disease, Respiratory Disease, Immune Disease, upper respiratory tract infection.

All 113 patients but one included in the study tested positive for COVID-19 by RT-PCR. The one patient was false negative on RT-PCR test and was confirmed COVID-19 positive based on CT-scan and serological test. Description of the patients including demography, pre-existing conditions, type and onset of symptoms is presented in Table 1. The mean (SD) age of the study participant was 45.9 (11.8) years and ranged from 20 to 67 years. 99 (87.6%) of the patients were men, and 70 (62.5%) were non-Emirati international residents (47 from Asia, 14 from Middle East, 4 from South America and 5 with no available data on detailed nationality). Nearly 80% of the patients showed symptoms at the time of admission. Additional baseline laboratory and serology characteristics are presented in S2 Table in S1 File.

Out of a total of 203 IgG measurements, six patients had repeated measurements within the same month and therefore 196 measurements were included in the monthly presentation. The monthly levels of IgG show a consistent increase during the first three months after the onset of symptoms with a peak level during the third month (Fig 2). Within the first 30 days after the onset of symptoms, the median (IQR) of IgG level was 45.4 (74.8) AU/ml. Out of 100 patients with IgG measurements within the first month, 69 (69.0%) showed a positive IgG response

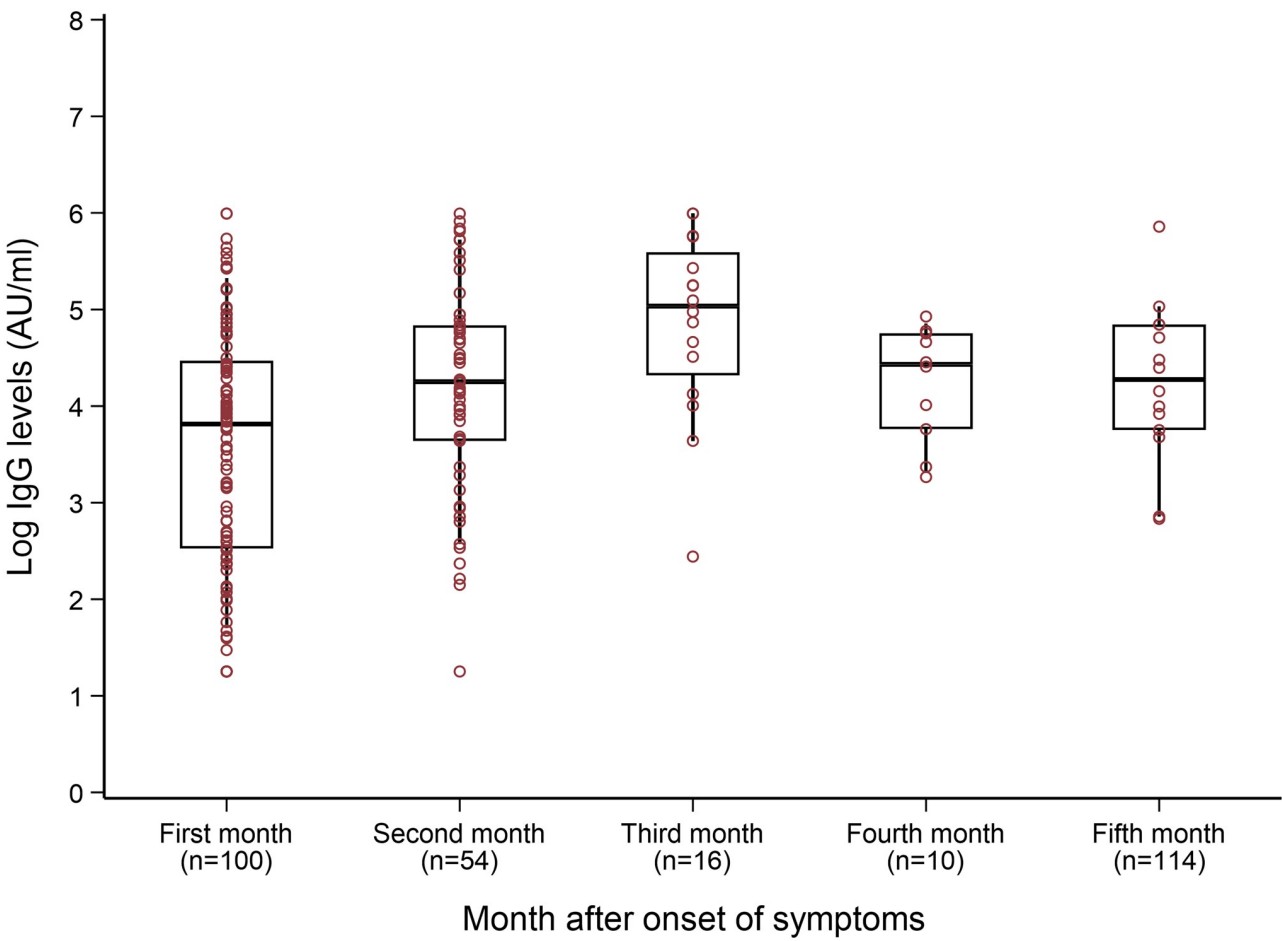

**Fig 2. IgG levels in the following months post-COVID-19 infection.** IgG- Immunoglobulin G measured as log to the base of 10. The boxes represent the distribution of the IgG levels in the patient population for different months, irrespective of the visits.

(≥ 15 AU/ml). The median (IQR) of IgG level increased to 70.3 (87.9) and 154 (195.5) AU/ml during the second and third month, respectively. Although the levels gradually decline afterward, all the patients kept levels greater than 27.7 and 17.4 AU/ml during the fourth and fifth month, respectively.

Fig 3 shows the graphical representation of the distribution of IgG levels (log 10 values) across the three visits. A positive IgG response has been observed from few days (first visit) to few weeks (second visit) to few months (third visit). The mean IgG level was different between visit 1 and visit 2 (P < 0.001), between visit 1 and visit 3 (P = 0.002) and no significant difference was found between visit 2 and visit 3. Similar trend was found when restricting the population to those having a complete follow-up (S1 Fig in S1 File). No difference was found between asymptomatic and symptomatic patients (S2 Fig in S1 File).

Table 2 shows the proportion of patients showing a positive IgG titer in the serology test. During the first serological measurements, there were 68.1% of patients who tested positive for IgG response. By the third visit, all the patients who reported back to the study had IgG antibody response. The baseline demographic, clinical, laboratory and serology characteristics of the patients by IgG response status are presented in S3-S6 Tables in S1 File. Patients who showed positive IgG response in visit 1 and visit 2 were older and had longer treatment durations compared to those showed negative response. Higher levels of Lactate Dehydrogenase

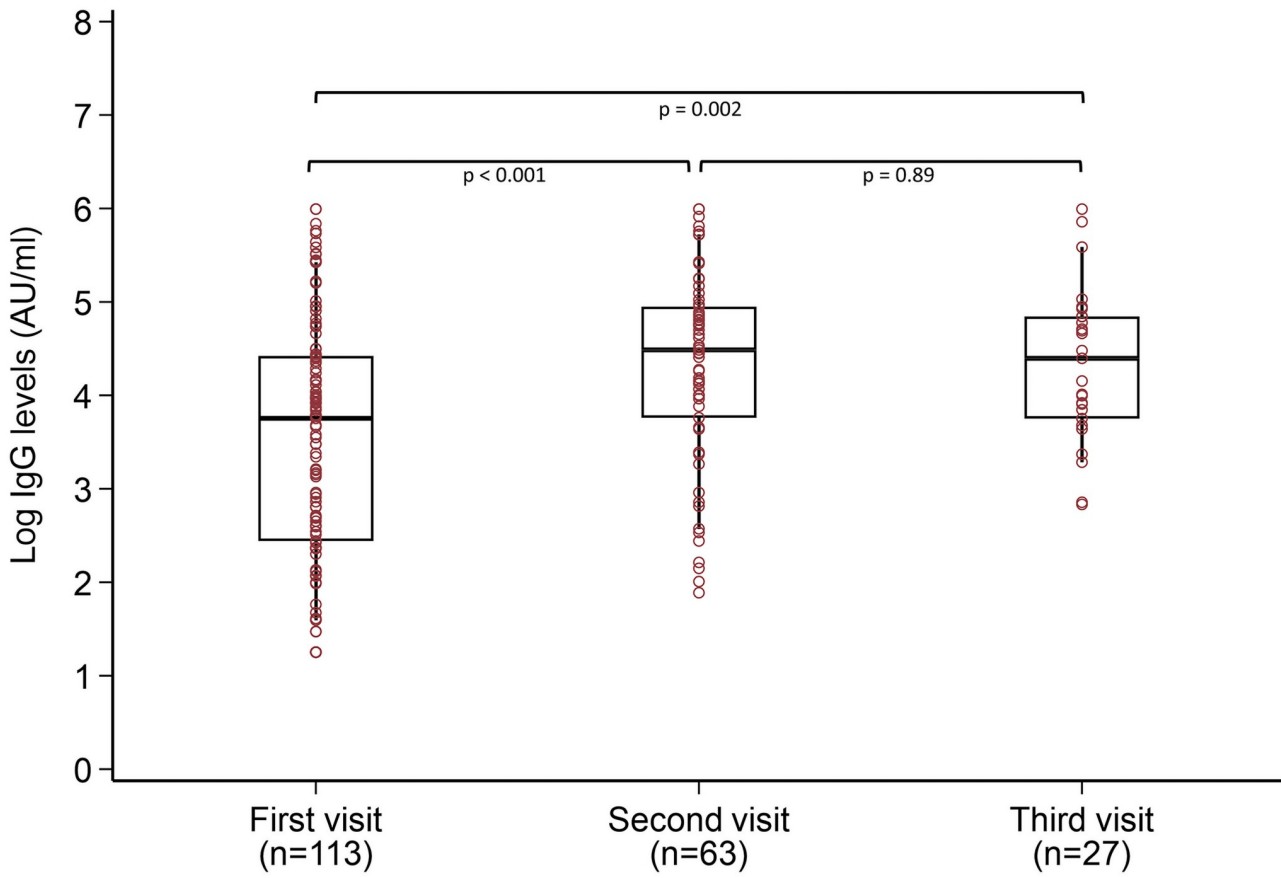

**Fig 3. IgG levels in the patients during the three visits.** Log- logarithmic value to the base 10. IgG- Immunoglobulin G. Each box plot represents the distribution of the IgG levels during different visits.

(LDH) and Ferritin were observed in patients with positive IgG response during visit 1, while lower levels of absolute lymphocyte count (ALC) and QTC2 were observed in patients with positive IgG response during visit 2 compared to those showed negative response. There were no statistically significant differences in the distributions of the other characteristics by IgG response status.

The differential trend in the IgG response in different patients is categorized into 'increasing' or 'decreasing' during their multiple visits. The increasing trend in the levels of IgG from the first to the second visit was observed in 82.5% of patients (Table 3). An increasing IgG levels from visit 2 to visit 3, and from visit 1 to visit 3 was observed in 55.6% and 70.4%. of the

**Table 2. Proportion of patients showing a positive IgG response during the study.**

| | Patient profile for IgG response | | |
|---|---|---|---|
| | First visit (n, %) | Second visit (n, %) | Third visit (n, %) |
| **Total number of patients** | 113 | 63 | 27 |
| **Positive IgG response** | 77 (68·1%) | 57 (90·5) | 27 (100) |
| **Negative IgG response** | 36 (31·9%) | 6 (9·5) | 0 (0) |

IgG- Immunoglobulin G.

**Table 3. Proportion of patients showing an increased IgG response between the visits.**

| | | N | All patients N (%) | Women N/N$_{total}$ (%) | Men N/N$_{total}$ (%) | Asymptomatic N/N$_{total}$ (%) | Symptomatic N/N$_{total}$ (%) |
|---|---|---|---|---|---|---|---|
| Visit 1 to visit 2 | | 63 | 52 (82.54%) | 8/8(100%) | 44/55 (80%) | 13/17 (76.47) | 39/46 (84.78) |
| Visit 2 to visit 3 | | 27 | 15 (55.56%) | 2/4 (50%) | 13/23 (56.52%) | 4/6 (66.67) | 11/21 (52.38) |
| Visit 1 to visit 3 | Any visit 3 | 27 | 19 (70.37%) | 3/4 (75%) | 16/23 (69.57%) | 5/6 (83.33) | 14/21 (66.67) |
| | Visit 3 after 90 days | 19 | 12 (63.16%) | | | | |
| | Visit 3 after 120 days | 14 | 9 (64.29%) | | | | |

IgG- Immunoglobulin.

patients, respectively. Furthermore, about 63.2% and 64.3% of the patients showed sustained increase in IgG response for more than 90 days and 120 days, respectively. Fig 4 maps the individual patient's trend in IgG response during the different time of visits. At the end of the study period, all patients continued to remain positive for IgG response. Reduction in the quantity of IgG from visit 1 to visit 3, was observed in 29.6% of the patients. Symptomatic patients have a higher increased IgG response between visit 1 and 2 than asymptomatic ones. However, because of the small sample size, no clear difference appears according to sex and presence of symptoms (Table 3).

The baseline demographic, clinical, laboratory and serology characteristics of the patients by IgG response status are presented in S7-S10 Tables in S1 File. Patients who showed increase in IgG levels between visit 1 and visit 2 were older than those who showed decrease in IgG levels. Patients who showed increase in IgG levels between visit 2 and visit 3 had shorter viral shedding time compared to those who showed decrease in IgG levels between the two visits.

The patients with three IgG level measurements did not differ from those having only one IgG measurement, except regarding nationality: the patients with complete follow-up were much more Non-national (85.6% vs. 55.8%, S11 Table in S1 File).

In univariate analyses testing the association between several factors and an increase in IgG response from first to second visit: only age, nationality and QTC3 were statistically significant and no statistically significant association was found in model adjusted on age (results not shown).

## Discussion

This study shows a sustained and prolonged positive immune response in COVID-19 recovered patients. None of the recovered patients were re-infected within the 6 months period of follow up. Around 69% of the patients developed positive IgG response within the first month after the onset of symptoms. The levels of IgG showed a consistent increase during the first three months after the onset of symptoms with a peak level during the third month. Although the levels gradually decline afterward, all the patients tested during the fourth and fifth month kept high positive levels of IgG.

Monitoring IgG response is an important aspect of understanding the scope of immunity development. There have been previous reports about the persistence of IgG against COVID-19 virus in particular [2,5,7–9]. However, the interpretation of such reported studies poses some limitations due to their retrospective nature, short duration, small sample size and lack of repeated quantitative estimation of IgG response. Our study is one amongst the very few prospective studies to confirm that there is a possibility for a sustainable antibody response probably attributable to strong immune memory. The serum samples were tested for the IgG response elicited specifically against viral spike proteins S1 and S2. The IgG response was

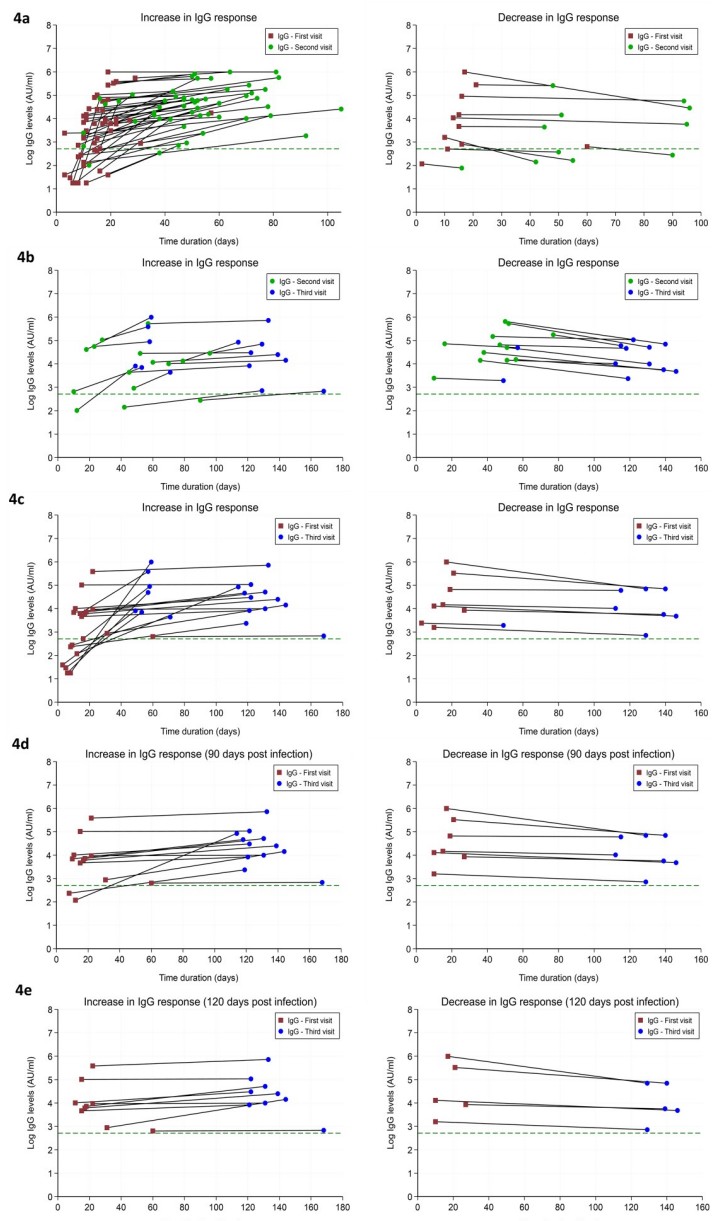

**Fig 4. Increasing and decreasing trend in IgG levels among patients during different visits.** Increasing and decreasing trend in IgG-Immunoglobulin (IgG) levels among patients during (a) first visit to second visit, (b) second visit to third visit, (c) first visit to third visit. Graphs (d) and (e) represent data of patients pertaining to first visit to third visit with IgG levels measured for more than 90 days and 120 days, respectively. The number of patient samples for first visit = 113, for second visit = 63, and for third visit = 27, respectively. Each individual lines represent the trend in IgG levels from one visit to the other. The green dashed line represents the minimum IgG level (15 AU/ml) beyond which the IgG response is considered to be positive. All IgG levels are represented on a logarithmic scale.

measured repeatedly using quantitative assay with high specificity for as long as 5.6 months in this study. A recent study from Iceland showed the possibilities of a continued antibody response post COVID-19 infections for at least a period of four months [9]. A study [2] in Wuhan has investigated the IgG and IgM response in COVID-19 patients for a period of 26 weeks. The study showed IgG positivity rate at about 70% at 6 months follow up period.

Another study among 87 American found a decrease in IgG level and memory B cell response consistent with antigen persistence, with a 6 months follow-up [17]. Among 210 Spanish patients, Pradenas et. al showed similar results in a study with a different antibody evolution according to severity of the disease [18]. They showed a slow decline in antibody level among mild/asymptomatic patients and a two-phase decline among hospitalized patients: fast initial decline, slowing down after 80 days [18]. Finally, a pre-print prospective study among 308 French patients that have found a half-live of anti-S IgG of 41 weeks, have shown that antibodies persist for longer periods of time in women than in men [19]. All these recent studies, published or pre-print have shown that despite a decay in humoral immunity there was a persistent humoral immunity after 5–7 months post-onset symptoms. Our results are concordant with these studies: although our numbers are small, we observed 100% of patients showing positive IgG response during the fourth and fifth months after the onset of symptoms. Our small sample size of women did not enabled us to compare our result with the French study regarding sex difference [19].

The quantitative nature of the study gives us an edge to monitor the trend in the individual patient's IgG response. About two-thirds of the patients had shown an increasing tendency of antibody response. Even the patients (29%) who showed a decreasing antibody response, remained positive for IgG levels (above 15 AU/ml). Further, our data shows the antibody levels keep increasing on a month-on month basis until the third month, with an inclination to reduce by the fourth and the fifth month. These results could be of value particularly, for designing the frequency of vaccination. The reduction in IgG levels during the fourth and fifth months could indicate a need to a booster dose of vaccination on a yearly basis.One of the limitations of our study is that we did not have data on IgG subclasses. Also, our IgG antibodies focused on S1 and S2 antigens only and no other isotypes. However, S protein is the main protein used as a target in COVID-19 vaccines and thus of very high interest.

The data collection was contingent upon the patient reporting to the hospital for the study duration. Out of the initial 113 patients, about 27 reported for an average of about 3–4 months with the maximum being 5.6 months. All these patients who reported till the end of the study had IgG levels higher than the cut-off limits for positivity. This significant outcome favors further developments to sustain, or to induce immunity through a potential vaccine. Vaccination could be a proactive measure to implement population- level immunity by administering it to the general populations, including those who were previously infected with COVID-19.

There are differences in the immune response of individuals to the virus, which opens door for scientific enquiry into the factors that predict such response and its magnitude and durability. Longer treatment durations and higher levels of LDH and Ferritin were observed in patients with positive IgG response which could indicate an effect of the severity of the disease on the immune response. Furthermore, older patients tend to have more positive IgG response and even increase in the IgG levels between the visits compared to younger patients; this may also reflect the fact that older patient are often the more severe ones. Moreover, similar to a previous report [3], serology results of patients with IgG antibody titers over 400 AU/ml, beyond the detection limit of the assay seems to be related with severity of the disease. In this cohort only four patients were admitted to ICU and three of them had IgG levels above 370 AU/ml, indicating a possible correlation between very high antibody levels and severity of the disease. Although this hint of an association between very high antibody titers and severity of the disease, the effects of other confounding factors like viral load, age, comorbidities, cell mediated immune response need to be explored thoroughly in future studies.

There are no reports suggesting an ethnic preponderance to COVID infection, although the disease behavior has been diverse [22,23]. The predictability of the general impact of the infection in an individual is further complicated by viral mutations, the immune status of the

individual, existence of comorbidities, age, sex among other factors. The multiethnic diversity of the UAE, demonstrated by having immigrant population constituting more than 87% of total population, according to mid-year 2019 estimate [22], provides a fundamental advantage of representing global population. Therefore, this study may be considered to be a valid representation of global population irrespective of ethnicity.

All the patients in this study were treated in accordance with the recommended treatment guidelines of the UAE [24]. The treatment included Hydroxychloroquine, Favipiravir, and a combination medicine containing Lopinavir and ritonavir. None of the patients were on steroids or any other immunosuppressive agents while we did not observe mortality in our study population during the entire study period.

The IgG antibody titers could be followed for a period of up to 6 months only in a small subset of sample (27 out of 113) owing to difficulties in obtaining serum samples due to the implementation of lockdown and access restrictions measures in the city during the study period. However, the patients were followed up over phone during the study period to ensure that there was no mortality in the study population and that they were free of COVID-19 infections. Furthermore, those who were followed up until the third visit did not seem to have different characteristics from those having had only one visit. A positive serology test result generally indicates previous exposure to the pathogen. Serologic results do not explicitly diagnose or exclude recent or past SARS-CoV-2 infection. Though less frequent, false positive results are possible due to prior infection with other coronaviruses. On the other hand, a negative immune response result may indicate absence or very low level of IgG antibodies to pathogen. This test could score negative in infected patients during incubation period and in early stage of infection leading to underestimation of the proportion of patients developed immune response. Moreover, an equivocal result should be interpreted with care as it may indicate a low level of IgG antibodies to pathogen in the sample or early stage of the response. Nevertheless, the repeated IgG measurements in this study enabled tracking the development process of the immune response and estimating the proportions of patients with increasing or decreasing immune response over time.

In conclusion, the study findings indicate a sustained and prolonged positive immune response in COVID-19 recovered patients. Vaccination against COVID-19 seems eminently possible and the chances of re-infection appears very low or none for at least within the first six months post infection. The developed immune response peaks during the third month after the onset of symptoms and gradually decline afterward, though remained positive. This would be a possible guide to the planning of booster doses for the to-be-used vaccines for COVID-19, including those who were previously infected with COVID-19.

## Supporting information

**S1 File.**
(DOCX)

## Acknowledgments

Al Zahrawi Medical services LLC provided reagents for our IgG testing.

## Author Contributions

**Conceptualization:** Ashraf Hassan Alzaabi.

**Data curation:** Khulood Al Mazrouei.

**Formal analysis:** Luai A. Ahmed.

**Investigation:** Abdulla E. Rabooy, Ali Al Zaabi, Mohammed Alkaabi, Falah AlMahmoud, Mai Farouk Hamed, Khalid Omar Bashaeb, Abdul Rahim Bakhsh, Suha Adil, Nadeen Elmajed, Ahmed Nigm Abousalha, Khulood Al Mazrouei.

**Methodology:** Ashraf Hassan Alzaabi, Khulood Al Mazrouei.

**Resources:** Abdulla E. Rabooy, Ali Al Zaabi, Mohammed Alkaabi, Falah AlMahmoud, Mai Farouk Hamed, Khalid Omar Bashaeb, Abdul Rahim Bakhsh, Nadeen Elmajed, Ahmad Kanaan Uwaydah, Khulood Al Mazrouei.

**Supervision:** Ashraf Hassan Alzaabi.

**Validation:** Ashraf Hassan Alzaabi.

**Visualization:** Khulood Al Mazrouei.

**Writing – review & editing:** Ashraf Hassan Alzaabi, Luai A. Ahmed, Khulood Al Mazrouei.

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
