## [Decision Letter · Decision Letter 0]

23 Mar 2021

PONE-D-20-39773

Longitudinal changes in IgG levels among COVID-19 recovered patients: A prospective cohort study.

PLOS ONE

Dear Dr. Alzaabi,

Thank you for submitting your manuscript to PLOS ONE. After careful consideration, we feel that it has merit but does not fully meet PLOS ONE’s publication criteria as it currently stands. Therefore, we invite you to submit a revised version of the manuscript that addresses the points raised during the review process.

All comments raised by Reviewer 1 and the additional editorial comments should be addressed in a revised manuscript.

We look forward to receiving your revised manuscript.

Kind regards,

Jishnu Das, Ph.D.

Academic Editor

PLOS ONE

Journal Requirements:

Additional Editor Comments:

Overall, the manuscript is technically sound and the data support the conclusions. However, all the points raised by Reviewer 1 need to be addressed. Points 1 and 2 are especially important to supporting the claims that a cutoff of 15 AU/ml is appropriate and the study includes a multi-ethnic population. In addition, the authors should also:

1. Discuss whether they expect the observed longitudinal changes to be the same for each IgG subclass (IgG1, G2, G3, G4) or different by subclass.

2. Explicitly describe the limitations of the study (only focuses on IgG and not other isotypes, only looks at IgG antibodies against S1 & S2 and not other antigens etc).

Reviewers' comments:

Reviewer's Responses to Questions

**Comments to the Author**

1. Is the manuscript technically sound, and do the data support the conclusions?

Reviewer #1: Yes

2. Has the statistical analysis been performed appropriately and rigorously? 

Reviewer #1: Yes

3. Have the authors made all data underlying the findings in their manuscript fully available?

Reviewer #1: Yes

4. Is the manuscript presented in an intelligible fashion and written in standard English?

Reviewer #1: Yes

5. Review Comments to the Author

Reviewer #1: The manuscript by Alzaabi et al describes a prospective cohort study of the longitudinal antibody response in adult COVID-19 patients in the UAE over a period of 6 months. Both symptomatic and asymptomatic patients were included within the cohort, and the many clinical parameters were recorded for each patient. The IgG response against SARS-CoV-2 S1 and S2 were determined using an FDA-EUA chemiluminescent immunoassay (CLIA) platform, and the authors find that across patients who returned for all three visits spanning 6 months, peak antibody titers against S1/S2 were observed at the second visit (approximately at month 3 post-symptom onset), and reduced slightly, but not significantly by the third visit (6 months post-symptom onset). They find that of those who returned for visit 3, all patients had developed humoral immunity.

Their data is consistent with reports from other countries regarding the longevity of the humoral immune response up to 6 months after symptom onset. The authors perform subsequent analysis between IgG seropositivity and clinical measures and find that seropositive patients at visit 1 had increased levels of LDH and ferratin compared with seronegative patients. The authors note trends in the data, where a majority of subjects have increasing IgG responses between visits, and maintained through day 120, and some patients show decreasing trends between visits. They show that symptomatic and asymptomatic patients do not differ in levels of IgG.

The authors present their study in a coherent and logical manner, and the inclusion of a comprehensive discussion of similar studies is welcome and places the author’s data in context. Their data set contributes to our understanding of long-term humoral immunity and against SARS-CoV-2 in populations around the globe.

Major points:

1. Were any control samples (pre-pandemic or non-COVID) included in the IgG analysis and used to establish the baseline cutoff of 15 AU/ml?

2. The authors mention that the patients represent a multi-ethnic and global population but only stratify the patients into Emerati national vs non-national. While the authors note that there were no differences in IgG across ethnicity/race, does that analysis refer to Emerati national vs non-national or to a more thorough breakdown and analysis of ethnicity/race?

3. It would be helpful if the authors could also plot the individual data points for the figures 2 and 3 in an overlay of the IQR plot, showing the variation across patients at each time point.

4. Figure 4 is a little confusing to me in the way it is currently presented, and although I understand the point that the authors are making with both Figure 4 and Table 3, I wonder if the maybe color coding each visit separately would help the readers correlate Figure 4 with Table 3 better. Another possibility may be to present all three visits on the same graph for all patients to give the readers a sense of which subjects responses are increasing/decreasing across the 6 months, or if there are some patients who are initially increasing, and then decrease. Also, it seems like the order of the graphs might make sense to switch to the 4th (Increase/Decrease in IgG response v1 vs v3) and 3rd graphs (Increase/Decrease over 90 days) to flow better with the text.

5. The number of clinical parameters that the authors collect on each patient represents a rich dataset for their COVID patients – does the anti-spike IgG response correlate with any clinical parameters in a regression analysis? A multivariate analysis approach of the data set may indicate parameters such as duration of viral shedding, treatments, or disease severity as drivers of IgG induction and maintenance. Along those lines, as the authors are able to divide patients into those whose titers increases vs those whose titers decrease, another interesting analysis would be if any of the clinical parameters could be used to predict increasing or decaying antibody responses. These analyses would help increase the relevance of the author’s work.

Minor comments:

1. Open parentheses missing in line 3

2. Line 22: suggest changing to “COVID-19 patients”.

3. Please include the dilutions (if any) of sera used in serology testing in the Methods.

4. Figure 4 is missing the labels for the graphs (a, b, c, etc.), and are labeled incorrectly as 2a, 2b, 2c etc. in the legend.

Reviewer: Bronwyn M. Gunn, PhD.

6. PLOS authors have the option to publish the peer review history of their article (what does this mean?). If published, this will include your full peer review and any attached files.

Reviewer #1: **Yes: **Bronwyn M. Gunn

---

## [Author Response · Author response to Decision Letter 0]

15 Apr 2021

Dear Reviewer,

Tank you for your pertinent comments. We have clarified some points and made the changes as you suggested . Please see below our point-by-point responses.

Best regards,

---

## [Editor Report · Decision Letter 1]

21 Apr 2021

Longitudinal changes in IgG levels among COVID-19 recovered patients: A prospective cohort study.

PONE-D-20-39773R1

Dear Dr. Alzaabi,

We’re pleased to inform you that your manuscript has been judged scientifically suitable for publication and will be formally accepted for publication once it meets all outstanding technical requirements.

Kind regards,

Jishnu Das, Ph.D.

Academic Editor

PLOS ONE
---

## [Editor Report · Acceptance letter]

24 May 2021

PONE-D-20-39773R1 

Longitudinal changes in IgG levels among COVID-19 recovered patients: A prospective cohort study. 

Dear Dr. Alzaabi:

I'm pleased to inform you that your manuscript has been deemed suitable for publication in PLOS ONE. Congratulations! Your manuscript is now with our production department. 

Kind regards, 

on behalf of

Dr. Jishnu Das 

Academic Editor

PLOS ONE